# Subgrouping patients with ischemic heart disease by means of the Markov cluster algorithm

Amalie D. Haue [1,2,9], Peter C. Holm[1,9], Karina Banasik [1], Kenny Emil Aunstrup[1], Christian Holm Johansen [1], Agnete T. Lundgaard [1], Victorine P. Muse[1], Timo Röder[1], David Westergaard [1], Piotr J. Chmura [1], Alex H. Christensen[2,3], Peter E. Weeke[2], Erik Sørensen [4], Ole B. V. Pedersen [4,5], Sisse R. Ostrowski [4,6], Kasper K. Iversen[3], Lars V. Køber [2,6], Henrik Ullum[7], Henning Bundgaard [2,5,10] ✉ & Søren Brunak [1,8,10] ✉

## Abstract

**Background** Ischemic heart disease (IHD) is heterogeneous with respect to onset, burden of symptoms, and disease progression. We hypothesized that unsupervised clustering analysis could facilitate identification of distinct and clinically relevant multimorbidity clusters.

**Methods** We included IHD patients who underwent coronary angiography (CAG) or coronary computed tomography angiography (CCTA) between 2004 and 2016 and used the earliest procedure as the index date. Patient health records were obtained from the Danish National Patient Registry, the Danish National Prescription Registry, and two in-hospital laboratory database systems. Genetic data were obtained from the Copenhagen Hospital Biobank. Using registered pre-index diagnosis codes ($n = 3046$), patients were clustered by application of the Markov Cluster algorithm. Multimorbidity clusters were then characterized using Cox regressions (new ischemic events, non-IHD mortality, and all-cause mortality) and enrichment analysis to explore both risks and phenotypical characteristics.

**Results** In a cohort of 72,249 patients with IHD (mean age 63.9 years, 63.1% males), 31 distinct clusters (C1-31, 67,136 patients) are identified. Comparing each cluster to the 30 others, seven clusters (9,590 patients) have significantly higher or lower risk of new ischemic events (five and two clusters, respectively). A total of 18 clusters (35,982 patients) have higher or lower risk of death from non-IHD causes (12 and six clusters, respectively), and 23 clusters have a statistically significant higher or lower risk for all-cause mortality. Cardiovascular or inflammatory diseases are commonly enriched in clusters (13). Distributions for 24 laboratory test results differ significantly across clusters. Polygenic risk scores are increased in a total of 15 clusters (48.4%).

**Conclusions** Based on prior disease profiles, unsupervised clustering robustly stratify patients with IHD in subgroups with similar clinical features and outcomes.

## Plain Language Summary

Ischemic heart disease (IHD) is among the leading causes of death world-wide. A major challenge is that the disease is highly heterogeneous and covers a wide range of different presentation forms, progression patterns and treatment responses. Despite this fact, patients diagnosed with IHD are commonly treated as one. In this study we sought to analyze patients diagnosed with IHD by identification of more homogenous subgroups. By describing patients with IHD with respect to all other pre-existing diseases they were diagnosed with, we identified subgroups that had different risk profiles and were characterized by different patterns when looking at their blood tests and genetic profiles.

Ischemic heart disease (IHD) is a common, chronic, and complex disease where the mode of onset, disease burden and disease progression are known to vary considerably between patients[1–3]. A major contribution to this heterogeneity relates to multimorbidity, as more than 85% of IHD patients have been diagnosed with other chronic diseases[4,5]. Although it is known that

multimorbidity tends to cluster, the increased mortality in patients with cardiometabolic multimorbidity is generally only related to single disease states, such as chronic obstructive lung disease (COPD), diabetes, or stroke[6]. Moreover, the risk of cardiovascular diseases is increased in many chronic, inflammatory disorders[7–9]. As the overall life expectancy in many countries

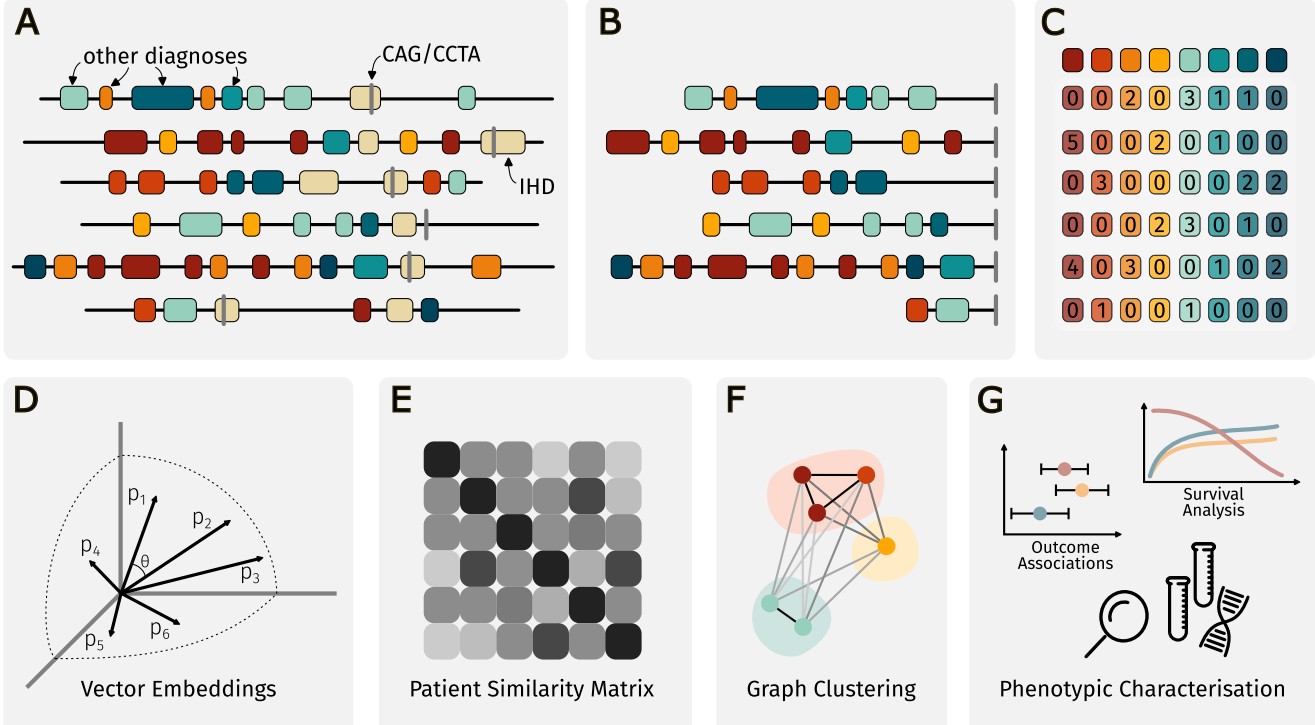

**Fig. 1 | Graphical overview of the study.** Steps **A–C**: Construction of patient-specific disease-frequency vectors by assembling the ICD-10 codes registered in the electronic health records. Using the date of the first CAG/CCTA as the index date, only ICD-10 codes registered before the index date are included. Steps **D** and **E**: Patient-specific vectors are embedded using SVD, and a patient similarity matrix is constructed from the cosine of the angle between the embedded vectors. Step **F**: Application of the MCL algorithm to obtain clusters of patients with specific patterns of multimorbidity. Step **G**: Characterization of the resulting clusters to examine their risk of three pre-defined outcomes and phenotypic characteristics defined from laboratory, medication and genetic data. CAG coronary angiography, CCTA coronary computed tomography angiography, ICD-10 International Statistical Classification of Diseases and Related Health Problems 10th Revision, IHD ischemic heart disease, MCL Markov Cluster, SVD singular value decomposition.

is increasing and comorbidities are accumulating, new methods for characterizing and studying cardiometabolic multimorbidity are needed[10–15].

Unsupervised clustering algorithms and other network-based methods can systematically reveal structure in large, feature-rich datasets and be used to identify distinct patient subgroups within a heterogenous population, leading to systematic characterisation of multimorbidity[6,16,17]. Proof-of-concept analyses of cardiovascular phenotypes, including IHD, heart failure, diabetes, and atrial fibrillation, have already been performed[18–24]. However, the study designs are rarely geared to detect signals in healthcare data in a data-driven manner[25].

For decades, Danish healthcare registries have had a strong position within epidemiological research[25–27]. We carried out an unsupervised clustering analysis of highly heterogeneous 72,249 patients with IHD based on their entire disease history until IHD onset. Explicitly, we classified IHD based on all pre-existing diseases covering the entire spectrum of multimorbidity. We identified 31 distinct patient subgroups derived from a pool of 3046 different diagnoses assigned prior to IHD onset. We quantitatively characterized all subgroups according to the strength of the associations with clinical outcomes (including survival), and identified which laboratory tests, medications, and genetic risk scores differed significantly across clusters.

## Methods
### Data sources and study population
Data from the Danish National Patient Registry (NPR) and the Danish Registry for Causes of Death were linked to in-hospital electronic health records (EHR) covering the two Danish healthcare regions in Eastern Denmark (~2.9 mil inhabitants), and the Copenhagen Hospital Biobank Cardiovascular Disease Cohort (CHB-CVDC)[26,28,29]. Linkage of different healthcare data sources was obtained via the personal identification number, and only patients admitted to a hospital in Eastern Denmark in the years 2004–2016 were considered[30]. We identified all patients in NPR who were assigned an ICD-10 code for IHD[31]. To increase the positive predictive value of IHD diagnoses and temporally align included patients, we further required that patients had been subjected to coronary arteriography (CAG) or coronary computed tomography angiography (CCTA) (Fig. 1). To qualify that CAG/CCTAs were conclusive for IHD, patients were only included if the CAG/CCTA was performed during a contact where patients were assigned an ICD-10 code for IHD. For a list of eligible ICD-10 and procedure codes, see Supplementary Table 1 in the Supplementary Information. We set the earliest CAG/CCTA fulfilling this criterion as the index date and excluded patients with an index date before the year 2004 or after 2016 (Fig. 2).

### Comprehensive analysis of diagnosis codes to map multimorbidity
Multimorbidity was assessed by constructing patient-level vectors enumerating level-4 ICD-10 diagnosis codes assigned up until the index date (Fig. 1). We excluded ICD-10 code for IHD (I20–I25); codes from chapters XV (pregnancy and childbirth), XVI (perinatal), XVII (congenital malformations and chromosomal abnormalities), XIX (injuries and poisonings), XX (external causes of morbidity and mortality), and XXI (administrative codes); and codes assigned to less than five patients (n = 1673). Thus, diagnosis codes from chapters I–XIV as well as chapter XVII (symptoms, signs and abnormal clinical and laboratory findings, not classified elsewhere) were included in the analysis. Using a simple "bag-of-words" approach, frequencies of included diagnosis codes (n = 3046) in each patient record were computed. The resulting 72,249 × 3046 count matrix

**Fig. 2 | Flowchart: data sources and study population.** NPR The Danish National Patient Registry. IHD ischemic heart disease (ICD-10 codes I20–I25). CAG coronary arteriography, CCTA coronary computed tomography angiography, ICD-10 International Statistical Classification of Diseases and Related Health Problems 10th Revision.

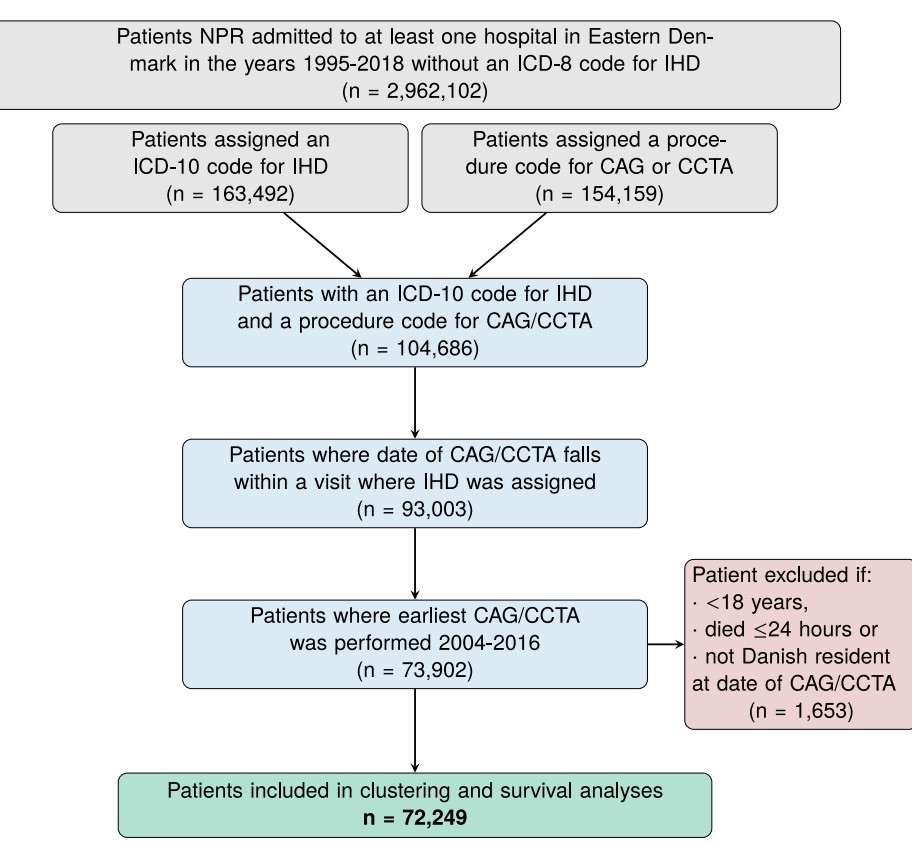

was used as a basis for the construction of a patient similarity network used as input for the Markov Cluster (MCL) algorithm[32].

### Construction of patient similarity network and network clustering by application of the Markov Cluster algorithm

To define the patient similarity network, a lower rank approximation of the diagnosis count matrix was created using the "truncatedSVD" implementation of singular value decomposition (SVD) from the Python package scikit-learn. We used the default fast "randomized" SVD algorithm, a fixed random seed of 42, and increased the number of iterations of the SVD solver to 10. The number of components in the output matrix was specified to 41, since we found that the accumulated explained variance of these was 50% (Supplementary Fig. 2).

To reduce the density of the patient similarity network, while still retaining an informative topology, all edges with an edge weight less than 0.35 were removed and the number of edges connected to each node were limited using the "#ceilnb" transformation from "mcl-edge"; a maximum of 8000 was used (Supplementary Fig. 2). The weights of the remaining edges in the network were shifted such that the lowest weight was 0.0 as recommended in the MCL manual[33]. The final pre-processed network was then used as input for the "mcl" implementation of the MCL algorithm. We used a pruning scheme of -P 7000 -S 800 -R 900 -pct 90, and a pre-inflation factor of 0.5 to make edge-weights more homogenous. For the clustering, we selected a pre-inflation parameter of 2.0 corresponding to the default in the MCL manual[34,35].

After clustering, we excluded any cluster with fewer than 500 individuals. The remaining clusters were denoted C followed by an integer indicating the rank of the clusters with respect to cluster size (number of patients in that cluster). Thus, C1 refers to the largest of the clusters. Cluster demographics were compared. Mean age at IHD onset in each cluster was compared to the mean age at onset in all the other clusters using Tukey's Honest Significant Difference (HSD) method. Significance level was set to 0.05, and P-values were adjusted using the Holm method assuming 465 tests (adj. P-val.).

### Robustness analysis

For cluster robustness assessment, diluted versions of the reference clustering were generated by deleting edges with a probability of $\alpha$, where $\alpha$ would range between 0 and 50%[36]. An $\alpha$ of 0 would leave the network unchanged. In contrast, the shuffled versions of the network had the same number of nodes and edges as the reference clustering. The shuffled networks were generated as described by Karrer et al.[37]. Finally, the generated clusters were compared to the reference clustering and the variances were quantified with reference to the so-called variation of information measure (VI)[36]. In total, 20 variations of the input graph were constructed by shuffling and deleting edges, respectively. Additionally, we also tested the impact of frequency scaling (term frequency-inverse document frequency, tf-idf) on the count matrix, as well as removing diagnosis codes assigned 3/6/12 months prior to the index date[38].

Clustering resulting from these different perturbations was compared with the original "reference" clustering, and their inter-cluster similarity was quantified using the split-join distance and the VI[39].

### Survival analysis

To investigate the association between cluster-membership and the competing risks of new ischemic events and death from non-IHD causes, and all-cause mortality, we used Cox proportional-hazards models (Cox models). The outcome "new ischemic event" was a composite outcome of (a) hospitalization minimum 30 days after index for myocardial infarction or unstable angina pectoris (i.e., hospitalization with myocardial infarction or unstable angina pectoris as the primary diagnosis), (b) revascularization not related to the index date, and (c) any death where IHD was listed as the primary or secondary cause. We applied the "elbow-method" to show that 75% of revascularizations occurred within 90 days of the index event, and therefore, these procedures were not considered secondary ischemic events. Similarly, 75% of subsequent hospitalizations for MI occurred within the first 30 days of the index event and were therefore not included in the endpoint. The unifying argument is that planned revascularization after the discovery of CAD

follows similar temporal patterns (Supplementary Fig. 1). Outcomes were obtained from NPR and the Danish Registry for Causes of Death. Eligible codes for inclusion, outcomes and specific cutoffs are available in the Supplementary Information.

Patients were followed from index until the occurrence of either of the three outcomes, or end of follow-up (year 2018), whichever came first. The dependent variable was either risk of new ischemic events, death from non-IHD causes, or all-cause mortality, and the independent variables were cluster, sex, and age at index. To age-adjust the models, analyses were performed using a restricted cubic spline with three knots for age at index. Follow-up time was truncated to a maximum of 5 years. For each cluster, hazard ratios (HRs) and 95% confidence intervals (CIs) were estimated by comparing HRs for the members of the cluster with those of non-members. Clusters associated with increased risk of all three outcomes were defined as "high-risk clusters".

## Preprocessing of laboratory and medication data

Clusters were characterized by laboratory, medication and genetic data based on the subset of patients where these data types were available.

The laboratory test results from the EHR data were originally archived in the administrative biochemical databases Labka and BCC[40]. In relation to the EHR data used in this study, Labka covers the hospitals with SHAK codes 1301, 1309, 1330, 1351, 1401, 1501, 1516 and 4001 in the Capital Region of Denmark and BCC covers the hospitals with SHAK codes 2000 and 2501 in Region Zealand for the periods 2009–2016 and 2012–2016, respectively (Supplementary Table 2). Biochemical laboratory tests were either classified in accordance with the Nomenclature, Properties and Units (NPU) or local systems[41]. Reference intervals were provided by the laboratories that analyzed the blood tests. Biochemical data were expected to be available for the patients where the index procedure was performed at a hospital located in either the Capital Region or Region Zealand at a time that was covered by the two databases.

A panel of 25 different lab parameters was included in the analyses. Only tests taken up to 90 days before the index or on the day of the index were included. Included lab tests were plasma levels of potassium, sodium, hemoglobin, estimated glomerular filtration rate (eGFR), creatinine, carbamide, glucose, troponin (I/T), HDL cholesterol, LDL cholesterol, total cholesterol, leukocytes, C-reactive protein, lymphocytes, monocytes, neutrophils, basophils, platelets, INR, alanine transaminase, albumin, alkaline phosphatase, bilirubin, and triglyceride.

A total of 48,957 patients (30,736 males and 18,221 females) were included from a hospital where biochemical data were available (67.8% of the entire cohort). As an indicator for data completeness and quality, the number of patients for whom available biochemical data at the time of index agreed with the clinical standard of care was assessed. This implied that patients had sodium, potassium, hemoglobin, and creatinine (or eGFR) measured a maximum of 90 days before or on the day of the index. The 31,224 patients who fulfilled this requirement were included in the biochemical analysis. Laboratory measurements available for at least 50% of these patients were included in the analysis. In cases where patients had more than one test available in the period from 90 days before to the index, the test closest to the index was used.

For every cluster, a *score* was computed based on the number of patients with a lab test below, within, or above the standard reference value, indicated by −1, 0 and 1, respectively. The *score* was defined as the mean of the summarized values (−1, 0 or 1) per cluster. Hierarchical clustering was then used to calculate the Euclidean distance between the *score* of each cluster for each test. All analyses of biochemical data were performed in R 3.6.2 using the "ComplexHeatmap" and "circlize" packages.

Medication data were obtained from the Danish National Prescription Registry, which contains data on all redeemed prescriptions from 1994 through 2023. Prescriptions are classified according to the WHO Anatomical, Therapeutic Chemical Classification codes (ATC-codes)[42]. Prescriptions were included in the enrichment analysis (see below) if they were redeemed within the period of 2 years, until 1 week before the index.

## Calculation of polygenetic risk scores for 14 traits

Polygenic risk scores (PGS) were calculated using the LDpred2 framework, implemented in the R package bigsnpr (v1.11.6) with R version 4.0.0 and the workflow management system Snakemake[43–45]. In preparation for PGS calculations, autosomal genotype data from 242,644 individuals in the CHB-CVDC[29] were filtered to only include variants present in LDpred2's recommended set of 1,054,330 reference variants. This recommended set is based on the reference set HapMap3 from the International HapMap project, which was established by genotyping 1.6 million single-nucleotide polymorphisms (SNPs) in 1184 individuals from 11 global populations[46]. Any missing genotype information was assumed to be the affected locus' reference allele.

We matched the remaining set of 994,643 genotyped variants with variants found in summary statistics data corresponding to 14 traits, obtained from nine GWAS meta-analyses (atrial fibrillation[47], BMI-adjusted type 2 diabetes[48], chronic kidney disease[49], HDL cholesterol levels[50], heart failure[51], LDL cholesterol levels[50], stroke[52], total cholesterol levels[50], triglyceride levels[50]) and five GWAS (acute myocardial infarction[53], coronary artery disease[54], diastolic blood pressure[55], non-alcoholic fatty liver disease[56], systolic blood pressure[55]). Variants present in both genotype and summary statistics data were then subject to LDpred2's recommended standard deviation quality control. After variant matching and quality control, a mean of 963,354 (S.D. 87,774) variants remained for subsequent per-chromosome risk score calculation for each of the 14 traits. We used the LDpred2-auto algorithm with 30 Gibbs sampling chains, 1000 burn-in iterations and 500 iterations after burn-in. The initial values for the 30 sampling chains were (a) the LDSC regression estimate for heritability $h^2$ (same for all chains); (b) one of 30 initial values for the proportion of causal variants $p$, evenly spaced on a logarithmic scale from $10^{-4}$ to 0.5.

Variant effect sizes were calculated from each set of 30 sampling chains (per trait and chromosome) through a three-step process, which serves to ensure that the model (spanning 30 chains) successfully converged: (1) computing the standard deviations of each chains' predicted scores, (2) keeping only the chains within three median absolute deviations from the median standard deviation, (3) averaging the effect sizes of the remaining chains. Across the 308 per-chromosome models (14 traits times 22 chromosomes), 28.9 chains were included in the final score on average. For each individual, we calculated per-chromosome risk scores by multiplying the average variant effect sizes by the individual's corresponding genotype, and then added the per-chromosome risk scores up into one genome-wide PGS. To ease comparisons across traits, each trait's PGS distribution was scaled to a mean of zero and a standard deviation of one.

## Enrichment analysis

The entire cohort as well as identified clusters were characterised by prevalence of diagnosis in each cluster, number and kind of redeemed prescriptions prior to index as well as laboratory values out of reference range and by genetics.

The phenotypic enrichment analysis was carried out based on ratios between observed (O) and expected (E) frequencies of diagnoses in the clusters (O/E-ratios). Ratios between the frequencies of ICD-10 codes in each cluster (observed frequencies) and the frequencies of ICD-10 codes in the entire population (expected frequencies) were calculated and expressed as O/E-ratios[57]. In subsequent characterization of clusters, enrichment denoted O/E-ratios > 2, and clusters were characterized as having little enrichment if the sum of the ten largest O/E-ratios < 50. Inverse changes were used to denote O/E-ratios between 0 and 1.

We defined patients with polypharmacy as cases with more than five different recurrent prescriptions each redeemed two or more times within 1 year prior to the index date[58]. Clusters where at least one laboratory value had been out of reference range within one year prior to the index were characterized as having skewed blood picture. Hierarchical clustering was also applied to estimate the cluster similarity with respect to the laboratory tests.

## Table 1 | Patient demographics, co-morbidities, and outcomes

| Cohort demographics | Total | Males | Females |
|---|---|---|---|
| *Cohort demographics* | | | |
| Number of patients (%) | 72,249 | 45,576 (63.1) | 26,673 (36,1) |
| Mean age at index (SD) | 63.9 (11.9) | 62.9 (11.6) | 65.6 (12.1) |
| *IHD manifestations (ICD-10)* | | | |
| Angina pectoris (I20) | 38,239 (52.9%) | 22,628 | 15,611 |
| Acute myocardial infarction (I21) | 33,299 (46.1%) | 27,720 | 10,579 |
| Subsequent myocardial infarction (I22) | 61 (<0.1%) | 34 | 27 |
| Certain current complications following acute myocardial infarction (I23) | 138 (<0.1%) | 92 | 46 |
| Other acute ischemic heart diseases (I24) | 1341 | 814 | 527 |
| Chronic ischemic heart disease (I25) | 22,750 | 14,589 | 8,152 |
| *Common comorbidities (ICD-10)* | | | |
| Primary (essential) hypertension (I10.9) | 24,818 | 14,508 | 10,310 |
| Hypercholesterolemia (E78.0) | 12,780 | 7842 | 4938 |
| Non-insulin-dependent diabetes (E11.9) | 7551 | 4891 | 2660 |
| Atrial fibrillation and atrial flutter, unspecified (I48.9) | 7075 | 4509 | 2566 |
| Heart failure, unspecified (I50.9) | 6160 | 4059 | 2101 |
| Chest pain, unspecified (R07.9) | 5863 | 3441 | 2422 |
| Senile cataract, unspecified (H25.9) | 5764 | 2795 | 2969 |
| Pneumonia, unspecified (J18.9) | 5469 | 3236 | 2260 |
| Hyperlipidaemia, unspecified (E78.5) | 5002 | 3306 | 1696 |
| Chronic obstructive pulmonary disease (J44.9) | 4621 | 2449 | 2172 |
| Common prescriptions (ATC) | Total | Males | Females |
| Simvastatin (C10AA01) | 19,088 | 11,710 | 7378 |
| Ibuprofen (M01AE01) | 15,595 | 9495 | 6100 |
| Acetylsalicylic acid (B01AC06) | 14,431 | 8791 | 5640 |
| Thiazides and potassium in combination(C03AB01) | 13,086 | 6280 | 6809 |
| Metoprolol (C07AB02) | 10,612 | 6130 | 4482 |
| *Outcomes, number of cases* | | | |
| New ischemic events (%) | 14,679 | 10,152 | 4,527 |
| Myocardial infarction | *5833* | *3709* | *2124* |
| Revascularization | *6282* | *4718* | *2124* |
| Death caused by IHD | *2563* | *1724* | *839* |
| All-cause mortality | 13,247 | 8434 | 4813 |
| Death from non-IHD causes (%) | 10,684 | 6710 | 3974 |
| Censored (%) | 33,639 | 20,299 | 13,359 |
| Outcomes, time to event µ (SD) | Mean time to event in years (SD) | | |
| | Total | Males | Females |
| New ischemic events | 1.48 (1.40) | 1.49 (1.41) | 1.48 (1.40) |
| Myocardial infarction | 2.40 (1.87) | 2.41 (1.89) | 2.38 (1.85) |
| Revascularization | 2.25 (1.88) | 2.28 (1.89) | 2.16 (1.84) |
| Death caused by IHD | 1.92 (1.13) | 1.95 (2.02) | 1.88 (2.05) |
| Death from non-IHD causes | 2.16 (1.50) | 2.14 (1.49) | 2.20 (1.51) |
| All-cause mortality | 3.08 (1.93) | 3.05 (1.94) | 3.13 (1.93) |
| Censored | 4.31 (1.11) | 4.29 (1.12) | 4.33 (1.09) |
| Total | 3.72 (1.64) | 3.67 (1.67) | 3.81 (1.60) |

*IHD ischemic heart disease, ICD-10 International Statistical Classification of Diseases and Related Health Problems 10th Revision, SD standard deviation.*

For each of the fourteen traits we calculated PRSs for, we used Wilcoxon rank-sum tests to compare the PRS distribution of each cluster to the combined PRS distribution of PRSs in all other clusters. Resulting *P*-values were converted to the false discovery rate (FDR) to account for multiple testing, with a total of 434 tests. We report effect sizes as calculated by the "wilcox.test" function built into R version 4.0.0. Level of significance was set to FDR < 0.05, assuming 434 tests.

### Statistics and reproducibility

The three sections, "Robustness analysis", "Survival analysis" and "Enrichment analysis" describe how the statistical analyses of the data were conducted,

### Ethics approvals and data access

The study was approved by The National Ethics Committee (1708829, 'Genetics of CVD'—a genome-wide association study on repository samples from Copenhagen Hospital Biobank), The Danish Data Protection Agency (ref: 514-0255/18-3000, 514-0254/18-3000, SUND-2016-50), The Danish Health Data Authority (ref: FSEID-00003724 and FSEID-00003092), and The Danish Patient Safety Authority (3-3013-1731/1/). In Denmark, research using registry data does not require specific ethical approval. The ethics committee has waived the requirement of informed consent for the use of genetic data in this project. Danish personal identification numbers were pseudonymized prior to any analysis. Study design, methods and results were reported in agreement with the STROBE statement[59].

### Reporting summary

Further information on research design is available in the Nature Portfolio Reporting Summary linked to this article.

## Results

### Cohort demographics and co-morbidities

A total of 72,249 patients (63.1% males, mean age 63.9 years) were included (Table 1). Angina pectoris (I20) was the most common IHD diagnosis (38,239 patients, 52.9%), followed by acute myocardial infarction (I21) (33,229 patients, 46.0 %) and chronic IHD (I25) (22,750 patients, 31.5%). A total of 68,103 patients (94.3%) had co-morbidities registered prior to the index. The most common diagnoses in the patient-level vectors were hypertension (I10.9) (24,818 patients, 34.4%), followed by dyslipidaemia (E78.0) (12,780 patients, 17.7%) and non-insulin-dependent diabetes (E11.9) (7551 patients, 10.5%). Prior to index, the mean number of diagnoses per patient was 8.1, and the mean number of redeemed prescriptions 2 years prior to index was 27.5. Prescriptions redeemed by most patients prior to index were Simvastatin, Ibuprofen, and Acetylic acid. A total of 14,679 patients experienced a new ischemic event during follow-up, and 13,247 died, of whom 10,684 patients died from causes other than IHD. Mean follow-up time was 3.72 years (Table 1).

### Identification and quantitative assessment of multimorbidity clusters

By application of the MCL algorithm to the patient similarity network, the algorithm identified 36 distinct multimorbidity clusters. The 36 clusters contained a total of 68,084 patients. Expectedly, the remaining 4365 patients (6.0% of included patients) were primarily patients with no diagnoses prior to index (>99%) and therefore appeared as singletons in the clustering. Clustering robustness was assessed as described in "Methods". We found that the shuffling of edges was most impactful on network topology in comparison to removing edges. The impact on network topology of removing edges 3 months prior to diagnoses was like that of shuffling 30% of edges (Supplementary Figs. 4 and 5). The primary care data (the Danish National Prescription Registry) was also used as a source of validation. Here we observed a positive correlation between cluster size and the number of redeemed prescriptions (Supplementary Fig. 6).

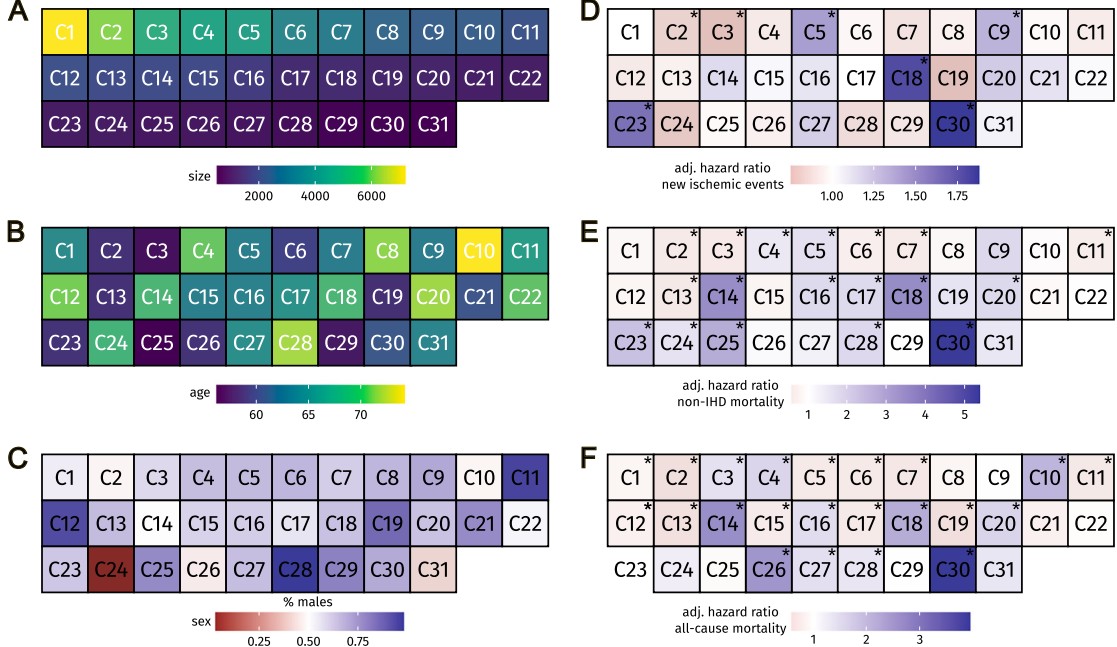

**Fig. 3 | Overview of quantitative characterization of multimorbidity clusters.** Each box in all panels represents a cluster (*n* = 31). In all panels, boxes are ranked by the number of patients in a specific cluster (decreasing going right and down). Yellow and green panels display cluster size (panel **A**) and mean age at index (panel **B**). Panel **C** displays the distribution of patient sex in each cluster. **C** Panels **D**–**F** display the dependent variables in the survival analyses. Panel **D**: New ischemic events. Panel **E**: Non-IHD mortality. Panel **F**: All-cause mortality. Asterisks mark statistical significance with adjusted *P*-value < 0.05. For exact *P*-values, see Table 2.

## Identification and quantitative assessment of multimorbidity clusters

Next, the 31 clusters that had >500 patients (67,136 patients) were characterized by demographics, clinical outcomes and phenotypic patterns (Fig. 3). The size of clusters ranged from 520 to 7191, and there were neither negative nor positive correlations between cluster size and mean age at index (Table 2). Using Tukey's HSD to compare the age at index between all 31 clusters (a total of 465 combinations), we found significant differences in 391 comparisons (84.1%, Supplementary Data 1). Distributions of patient sex in clusters also differed remarkably, where the proportion of males in clusters ranged from <25% to >75% (Fig. 3). Taken together, patterns picked up by the MCL algorithm are indicative of important qualitative differences between patient subgroups in IHD. For demographics of singletons and clusters of size <500, see Supplementary Table 3.

## Association of multimorbidity clusters with clinical outcomes

Risks for new ischemic events, death from non-IHD causes, and all-cause mortality in each of the 31 clusters were compared to the pooled risk for patients in the remaining 30 clusters (Fig. 3). Comparing each cluster (*n* = 1) to all the others (*n* = 30), seven clusters (20,221 patients) had a statistically significantly higher or lower risk of new ischemic events (Adj. *P*-val. < 0.05). A total of 23 clusters had a statistically significantly higher or lower risk for all-cause mortality. All clusters at increased risk of death from causes other than IHD were also associated with increased risk of all-cause mortality. Three clusters (C1, C15, and C17) associated with a reduced risk of all-cause mortality. None of these clusters associated with any of the other outcomes (Adj. *P*-val. < 0.05). Five clusters (9590 patients) and two clusters (10,631 patients) were at increased and decreased risk of new ischemic events, respectively. Similarly, 18 clusters (43,173 patients) had a statistically significantly higher or lower risk of death from non-IHD causes (Adj. *P*-val. < 0.05); where 14 clusters (21,282 patients) and four clusters (21,891 patients) were at increased or decreased risk of death from non-IHD causes, respectively. All clusters at increased risk of new ischemic events, associated with the risk of death from non-IHD causes as well. The same was true for the two clusters at decreased risk of new ischemic events, i.e., these clusters were also at decreased risk of death from non-IHD causes. Interestingly, there were 12

clusters at increased risk of death from non-cardiac causes and all-cause mortality, respectively. Among these, 11 clusters associated with increased risk of both death from non-IHD causes and all-cause mortality. Cluster C24 associated with an increased risk of death from non-IHD causes only, whereas cluster C27 associated with an increased risk of death for all-cause mortality only. A total of 13 clusters (23,963 patients) did not have an altered risk of the two outcomes, when compared to the other clusters (Table 2).

## Signals captured by the MCL algorithm and integration with other data types

For patients in the 31 clusters, we had laboratory measurements on 30,755 (49.5%) and genetic data on 19,422 (31.3%). To assess if the phenotypic differences captured by the MCL algorithm were also reflected in laboratory measurements, we tested if the distributions of test results within and out of reference ranges differed significantly. There were significantly different distributions of tests within and out of reference ranges in clusters for the 24 most frequent tests. Overall, this indicates that the phenotypic patterns within the entire spectrum of cardiovascular multimorbidity registered before the index correlate with results of clinical laboratory tests (Supplementary Table 4). Thus, these findings are a strong indicator that the patterns captured by the MCL algorithm are biologically relevant.

Figure 4 displays a summary of the results of the enrichment analysis. Overall, clusters enriched for diabetes and diabetes complications (C5, C23, C18, and C30) associated with increased risk of all three outcomes. These clusters were also characterized by polypharmacy and a high degree of laboratory tests that were out of reference range. In contrast, clusters that did not associate with risk of secondary ischemic events (C1, C6, C15, and C17) were not characterized by polypharmacy (Fig. 4).

## Enrichment analysis differentiates clusters with similar risk profiles

High-risk clusters were enriched for diabetes, peripheral atherosclerosis, cardiomyopathy and chronic kidney disease, respectively. The MCL algorithm also mapped signals in diagnoses that are known to co-occur, such as complications of diabetes (e.g., renal complications). Three of the five high-

**Table 2 | Cluster demographics, characteristics, and associations with outcomes**

| Cluster | Size | Mean age at index in years (SD) | Males | Fe-males | New ischemic events | | Death from non-IHD causes | | All-cause mortality | |
|---|---|---|---|---|---|---|---|---|---|---|
| | | | | | HR | Adj. *P*-val. | HR | Adj. *P*-val. | HR | Adj. *P*-val. |
| C1 | 7191 | 64.8 (11.3) | 3897 | 3294 | 1.000 | 1 | 0.856 | 0.103 | 0.863 | **2.74e$^{-4}$** |
| C2 | 5990 | 58.6 (11.5) | 2862 | 3127 | 0.825 | **7.24e$^{-5}$** | 0.600 | **1.73e$^{-10}$** | 0.645 | **0** |
| C3 | 4641 | 56.8 (11.4) | 2727 | 1914 | 0.757 | **3.25e$^{-8}$** | 0.586 | **2.04e$^{-8}$** | 0.653 | **2.07e$^{-14}$** |
| C4 | 4401 | 69.6 (10.2) | 2853 | 1548 | 0.920 | 1 | 1.461 | **0** | 1.403 | **0** |
| C5 | 4290 | 63.9 (10.7) | 2803 | 1487 | 1.402 | **0** | 1.629 | **0** | 1.571 | **0** |
| C6 | 3589 | 59.7 (10.9) | 2.388 | 1201 | 0.969 | 1 | 0.675 | **2.14e$^{-4}$** | 0.732 | **1.67e$^{-7}$** |
| C7 | 3309 | 63.8 (11.0) | 2025 | 1284 | 0.889 | 1 | 0.611 | **1.53e$^{-7}$** | 0.690 | **5.52e$^{-11}$** |
| C8 | 2802 | 71.1 (10.9) | 1867 | 935 | 0.943 | 1 | 0.842 | 0.551 | 0.866 | 0.051 |
| C9 | 2581 | 63.7 (11.8) | 1803 | 778 | 1.314 | **1.34e$^{-7}$** | 1.789 | **0** | 2.010 | **0** |
| C10 | 2562 | 74.2 (9.6) | 1225 | 1337 | 0.978 | 1 | 0.928 | 1 | 0.986 | 1 |
| C11 | 2292 | 66.1 (11.0) | 2186 | 106 | 0.926 | 1 | 0.650 | **4.89e$^{-5}$** | 0.716 | **8.04e$^{-8}$** |
| C12 | 2213 | 70.3 (10.2) | 2068 | 145 | 0.920 | 1 | 0.805 | 0.241 | 0.815 | **1.57e$^{-3}$** |
| C13 | 2070 | 58.6 (10.2) | 1348 | 722 | 0.946 | 1 | 0.577 | **1.21e$^{-3}$** | 0.599 | **3.40e$^{-8}$** |
| C14 | 2070 | 68.2 (9.6) | 1030 | 1010 | 1.146 | 1 | 3.390 | **0** | 2.566 | **0** |
| C15 | 2040 | 63.9 (10.1) | 1208 | 805 | 1.031 | 1 | 0.784 | 0.877 | 0.745 | **3.11e$^{-4}$** |
| C16 | 1654 | 64.1 (12.1) | 1013 | 641 | 1.107 | 1 | 1.761 | **1.86e$^{-13}$** | 1.483 | **2.54e$^{-12}$** |
| C17 | 1281 | 65.3 (9.9) | 714 | 567 | 1.001 | 1 | 1.761 | 0.299 | 0.735 | **7.24e$^{-3}$** |
| C18 | 1251 | 68.2 (9.8) | 802 | 449 | 1.790 | **0** | 3.421 | **0** | 2.129 | **0** |
| C19 | 1168 | 58.5 (9.7) | 995 | 173 | 0.752 | 0.085 | 1.571 | 0.146 | 0.616 | **8.42e$^{-4}$** |
| C20 | 1119 | 71.5 (11.3) | 713 | 406 | 1.213 | 0.421 | 1.782 | **1.45e$^{-13}$** | 1.576 | **0** |
| C21 | 1000 | 61.0 (11.0) | 769 | 231 | 1.116 | 1 | 0.890 | 1 | 0.806 | 1 |
| C22 | 988 | 69.2 (10.4) | 516 | 472 | 1.023 | 1 | 0.978 | 1 | 0.977 | 1 |
| C23 | 935 | 58.7 (12.2) | 588 | 347 | 1.609 | **7.59e$^{-10}$** | 2.275 | **0** | 1.951 | **0** |
| C24 | 932 | 67.9 (10.1) | 28 | 904 | 0.787 | 1 | 1.589 | **1.22e$^{-4}$** | 1.256 | 0.146 |
| C25 | 860 | 56.2 (9.9) | 664 | 196 | 0.978 | 1 | 2.691 | **0** | 2.461 | **0** |
| C26 | 852 | 58.7 (12.1) | 391 | 461 | 0.939 | 1 | 1.108 | 1 | 0.943 | 1 |
| C27 | 823 | 65.1 (10.9) | 532 | 291 | 1.201 | 1 | 1.289 | 1 | 1.430 | **4.48e$^{-5}$** |
| C28 | 686 | 71.7 (8.0) | 673 | 13 | 0.866 | 1 | 1.786 | **1.48e$^{-8}$** | 1.345 | **3.63e$^{-3}$** |
| C29 | 550 | 57.2 (11.1) | 435 | 115 | 0.906 | 1 | 0.985 | 1 | 0.986 | 1 |
| C30 | 533 | 61.2 (11.7) | 391 | 172 | 1.874 | **1.22e$^{-10}$** | 5.364 | **0** | 3.951 | **0** |
| C31 | 520 | 64.4 (11.2) | 213 | 307 | 1.052 | 1 | 1.484 | 0.370 | 1.356 | 0.130 |
| NA* | 5113 | 60.1 (11.1) | 3878 | 1235 | NA | NA | NA | NA | NA | NA |

*Patients that did not cluster or were in clusters of size <500. Bold indicates statistical significance. *N* = 31. *SD* standard deviation, *HR* hazard ratio.

risk clusters were also characterized by polypharmacy, whereas only C30 was enriched for a symptom (R50.9: fever, unspecified). All high-risk clusters were also characterized by having multiple laboratory tests out of reference range, consistent with pre-existing diseases affecting multiple organs (Fig. 4). Detailed results from the enrichment analysis are available in Supplementary Data 2.

Clusters enriched for aortic stenosis (C20) and stroke (C27) did not associate with increased risk of new ischemic events. As expected, the cluster enriched for aortic stenosis associated with increased risk of all-cause mortality. This association was also reflected in the laboratory tests, with a pattern like the ones identified among high-risk clusters. Interestingly, the cluster enriched for stroke did not associate with any of the pre-specified outcomes. Both clusters were characterized by polypharmacy. Consistent with the standard of care, the cluster enriched for stroke was also characterized by dual-antiplatelet therapy (Fig. 4).

Overall, clusters that did not associate with altered risk of new ischemic events displayed three different patterns. Clusters C1, C6, and C15, and C17 were all characterized by enrichment of known risk factors for the development of IHD, i.e., hypertension and hypercholesterolemia. None of these clusters was associated with an altered risk of all-cause mortality. Cluster C6 was not characterized by polypharmacy, as only a high fraction of patients had redeemed prescriptions for statins. This cluster also associated with decreased risk of death from non-IHD causes. In contrast, clusters C1, C15, and C17 were characterized by polypharmacy. None of the clusters characterized by well-known risk factors for the development of IHD had trends in the biochemical profiles that were indicative of multi-organ disease. Taken together, these observations are consistent with the expected effects of primary and secondary prevention.

**Clusters and their association with genetic data**
Finally, we identified 41 cases (out of 434 tests) where the PRS distribution for a specific trait in a cluster was significantly different from that trait's combined PRS distribution of the other 30 clusters. Among these cases, we found the largest effect size to be a higher genetic risk for atrial fibrillation in cluster C4 (0.57, FDR < 0.001) as well as a higher genetic risk for non-insulin-dependent diabetes in cluster C5 (0.55, FDR < 0.001). These findings are congruent with the results of the enrichment analysis for C4 and C5,

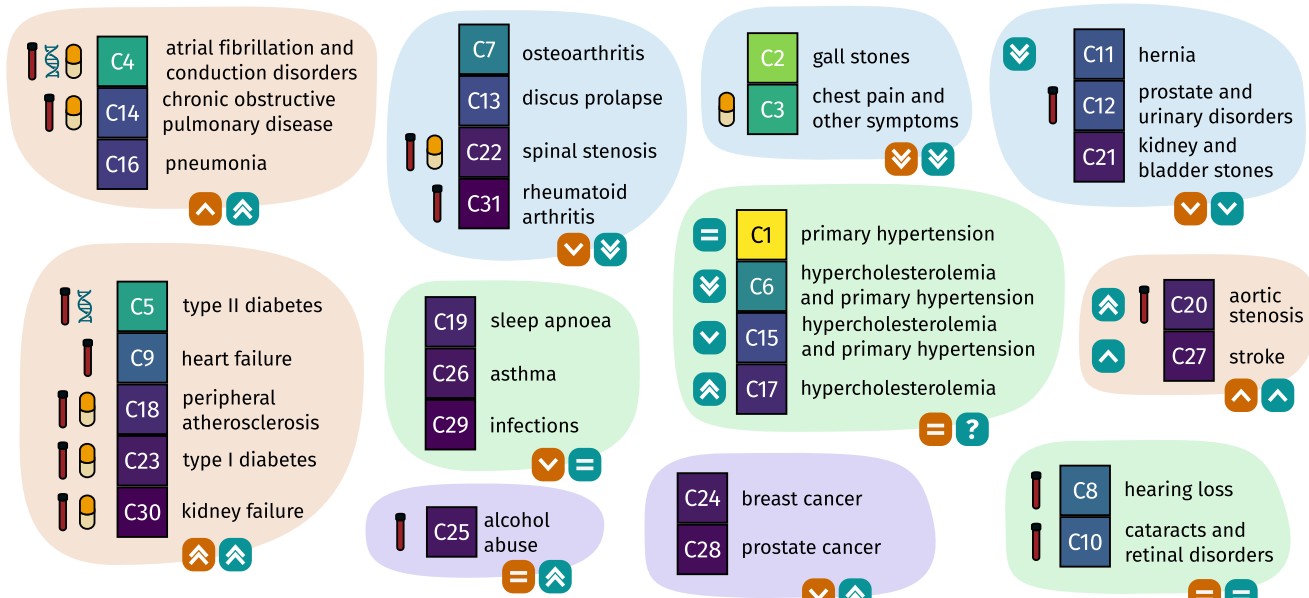

**Fig. 4 | Graphical summary of study results.** Boxes represent clusters, and clusters are grouped according to similarity with respect to results from the survival analysis. Boxes located bottom right of all groups represent the results from the survival analysis. Orange boxes: Risk of secondary ischemic events. Green boxes: Risk of death from non-IHD causes. Symbols to the left of the boxes that represent clusters indicate characteristic findings from integrated analysis of medication data, laboratory test results, and genetic data, respectively. IHD ischemic heart disease.

respectively. In contrast, C1 (phenotypically characterized by inverse changes) had relatively large, positive effect sizes for systolic as well as diastolic blood pressure (0.20 and 0.16, FDR < 0.001). Similarly, there were positive effect sizes for total cholesterol and triglycerides in C6, which was also characterized by little phenotypic enrichment as well as a high degree of inverse changes. A list of significant effect sizes for the 41 significant cases is shown in Supplementary Table 5.

## Discussion

Multimorbidity is a recognized and growing clinical challenge, where a systematic approach is currently lacking from clinical practice[5,6]. Secondary use of electronic healthcare data could address the challenges of patient heterogeneity by mapping clinically relevant patient subgroups, taking advantage of the vast amount of information that has been excluded from traditional observational studies. Due to its complexity, bottom-up, mechanistic approaches to obtain an actionable understanding of disease interactions are not likely to be immediately feasible, prompting the secondary use of electronic healthcare data for mapping patient heterogeneity and creating clinically relevant patient subgroups. Further, a general shift in the conceptual framework of coronary artery disease (an anatomical conception of IHD) was recently advocated, as current approaches and funding strategies focus on atherosclerotic plaque rupture or erosion rather than earlier disease stages[60]. Such earlier disease stages are likely to vary between patient subgroups owing to the impact of multimorbidity on disease onset and progression.

This study was designed to analyze population-wide, feature-rich data, and the number of multimorbidity clusters was not pre-specified. Previous studies that have explored the potential of clustering analysis within the cardiovascular domain have typically been limited to include up to 20 features[22,23]. As such, this study is among the first to analyze multimorbidity among patients with IHD in a deep and non-hypothesis data-driven manner without deciding up-front on the number of clusters. By means of the robustness analysis, we provide evidence that patients cluster into meaningful subgroups. Interestingly, these subgroups may display quite different survival and outcome characteristics, even if two clusters are relatively close in terms of patient similarity. When distinct quantitative differences in disease manifestation prior to the index date are handled non-linearly by the clustering framework, the non-uniform outcome distributions can be reflected. For example, we find that the mortality is not always driven by IHD. For example, the cluster enriched for COPD (C14) associates with an increased risk of death from non-IHD causes, but not new ischemic events (Table 2). No matter how aggressively one would treat these patients for IHD, it would likely be the comorbidity that would determine their survival. The clustering also provides a basis for preparing clinical trials that take the heterogeneity of the trial patient population into account. The clustering method can be used to perform a more fine-grained inclusion of patients, which would then lead to larger effect sizes.

Prior studies have handled prior disease in a much more hypothesis-driven manner.

Forman et al. defined multimorbidity as 'two or more medical diseases or conditions, each lasting more than one year'[4]. Similarly, the study using 20 features limited their definition to only cover the most common chronic conditions[23]. However, since we accrued the number of admissions for each diagnosis, the chronic nature of certain conditions is likely implicitly accounted for. In contrast to cluster analysis centered on patients with atrial fibrillation[20,24], the present study does not assess the association of clusters with a treatment outcome. Future cluster analyses of IHD that associate with treatment outcomes will be key in establishing evidence for symptom-based treatment. This perspective is of increasing clinical interest as mortality rates are declining, meaning that evidence of effects within traditional study designs and endpoint sare also increasingly difficult to establish[1]. Further, this perspective is in line with the choice of including risk of secondary ischemic events and death from non-IHD causes in the survival analyses. These outcomes also mark an original attribute of the study. Moreover, the clustering emphasizes patient groups where overrepresented comorbidities associate with relevant clinical outcomes.

The overall argument that motivated this study, is that graph theory and the application of unsupervised clustering to patient record data is a valuable approach to obtain a better understanding of IHD[11,17].

Five limitations of the study are (1) that laboratory and genetic data were not available for all patients (missingness), (2) that the temporal order of the prior diseases was not considered in the clustering, (3) that a cohort of patients with IHD identified in nationwide Danish health data will consist mainly of Caucasians, (4) the design does not allow for inclusion of asymptomatic IHD patients, and (5) the quality of the coding.

The first limitation was addressed by developing post-hoc methods for the assessment of cluster robustness and conventional analysis of the underlying data structures. The risk of introducing bias was limited, as no manual feature selection was exerted prior to clustering. Incorporating the chronological order of the diagnoses in the patient vectors would allow for a more nuanced description of the co-morbidity burden of the individual patient and could, in addition, help alleviate the limitation of chronic versus acute conditions. It is also possible that an even larger number of clusters would result, reducing the clarity of the IHD overview provided. Diseases are not always diagnosed in the order they appear; again, this creates noise in the overall characterization of IHD. The third limitation requires conducting a similar study in a different cohort. Importantly, the fact that the routine care data are being analyzed as is, rather than collected in a controlled setting, means that the results better reflect the actual IHD cohort. The choice of Cox modelling in time-to-event analysis can also be debated, but its limitations are of minor relevance as our goal was to design a method for conducting explorative studies, and therefore not a prediction study. An alternative would be the Fine-Gray subdistribution analysis, which was not applied, as this would be more relevant in the context of prediction. The fourth limitation is related to the use of routine care electronic patient records rather than screening data. Therefore, asymptomatic IHD is notoriously difficult to capture. Even if it were attempted, it would likely be a highly biased subgroup within asymptomatic IHD that could be added to the dataset. The fifth limitation concerns the quality of the coding that is often debated across settings, as well as the bias that is notoriously present in any routinely collected dataset. It is crucial to be aware of the fact that the clustering was performed on secondary care data. All patients had their diagnosis of IHD confirmed either by CAG or CCTA. The ICD-10 codes from the Danish registries used as input for the MCL algorithm have been extensively validated in the literature[26,30,31]. Despite this fact, differences in coding practices as well as thresholds for performing diagnostic procedures might vary between different countries, which again may impact the generalisability of the results presented in this study. Be that as it may, the results demonstrate the importance of capturing the entire spectrum of comorbidities that in all countries coexist in this patient subgroup. Further, by analysing the entire spectrum of comorbidities, the risks of introducing biases during data preprocessing are limited.

In addition, we would like to stress that the purpose of the study was to map the diversity in IHD comorbidity, highlighting differences in progression and risks rather than benchmarking clustering algorithms. All clustering algorithms have run-time dependencies, and it is not feasible to get those run-time parameters fully out of the equation. Minor differences in clustering results would likely not reflect IHD heterogeneity but rather algorithmic differences. By preparing the data taking IHD onset into account, we have provided a much more robust starting point for the clustering.

## Conclusion

In sum, the study showcases the strengths of a more fine-grained analysis of patient subgroups, which, in turn, may pave the way for successful implementation of precision medicine. Owing to its flexibility, the comprehensive, data-driven analysis of cardiovascular multimorbidity represents a novel method for characterizing multimorbidity in IHD with great potential for applying it to other diseases of interest where the overall disease burden is complex and heterogeneous.

## Data availability

Application for registry data access can be made to the Danish Health Data Authority (contact: servicedesk@sundhedsdata.dk). Anyone wishing access to the data and using them for research will be required to meet research credentialing requirements as outlined at the authority's web site: sundhedsdatastyrelsen.dk/da/english/health_data_and_registers/research_services. Requests are normally processed within three to six months. The source data for Fig. 3 are available in Supplementary Data 3. Source data for Fig. 4 are available in Supplementary Data 2 and Supplementary Table 5.

## Code availability

The code used to generate the results, including the clustering pipeline, is publicly available via https://doi.org/10.5281/zenodo.15211110[35].

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

## Acknowledgements

This work was financially supported by Novo Nordisk Foundation (Grants NNF17OC0027594 and NNF14CC0001) and the Innovation Fund Denmark via the NordForsk project PM Heart (5184-00102B). The authors would like to thank (1) research programmer, Troels Siggaard, Novo Nordisk. Foundation Center for Research, University of Copenhagen, Denmark, for continuous and reliable infrastructure support, and (2) Head of Cardiovascular Research, Hilma Hólm, deCODE genetics, Icelan,d for insightful comments.

## Author contribution

A.D.H.: conceptualization, methodology, software, formal analysis, investigation, visualization, writing—original draft, writing—review & editing. P.C.H.: conceptualization, formal analysis, data curation, writing— original draft, writing—review & editing. K.B., A.T.L., T.R.: methodology, writing—original draft, writing—review & editing. K.E.A., C.H.J., V.P.M.: software, formal analysis. D.W.: Supervision. P.J.C.: data curation and management. A.H.C., P.E.W.: writing—review & editing. E.S., O.B.V.P., S.R.O.,: data curation and funding acquisition. K.K.I., L.V.K., H.U.: supervision, writing—review & editing. H.B.: conceptualization, methodology, resources, funding acquisition, project administration, supervision, writing—original draft, writing—review & editing. S.B.:

conceptualization, funding acquisition, project administration, supervision, writing—original draft, writing—review & editing.

## Competing interests

S.B. received personal compensation for managing board membership at Intomics and Proscion and is a scientific advisory board member of Biocenter Finland, Health Data Research UK, the Finnish Center of Excellence in Complex Disease Genetics, ELIXIR Node (Luxembourg), Lund University Diabetes Centre (Lund, Sweden), and SciLifeLab (Stockholm, Sweden). S.B. reports stocks in Intomics, Hoba Therapeutics Aps, Novo Nordisk, Elly Lilli & Co., and Lundbeck. H.B. has received lecture fees from Amgen and Bristol-Myers Squibb. L.K. is a member of the speaker steering committee for Astra-Zeneca and Novartis. L.K. has received lecture fees from Novo Nordisk and Boehringer Ingelheim. The remaining authors declare no competing interests.

## Additional information

¹Novo Nordisk Foundation Center for Protein Research, Faculty of Health and Medical Sciences, University of Copenhagen, Copenhagen, Denmark. ²Department of Cardiology, The Heart Center, Rigshospitalet, Copenhagen, Denmark. ³Department of Cardiology, Copenhagen University Hospital, Herlev, Denmark. ⁴Department of Clinical Immunology, Copenhagen University Hospital, Copenhagen, Denmark. ⁵Department of Clinical Immunology, Zealand University Hospital, Køge, Denmark. ⁶Department of Clinical Medicine, University of Copenhagen, Rigshospitalet, Copenhagen, Denmark. ⁷Statens Serum Institut, Copenhagen, Denmark. ⁸Copenhagen University Hospital, Rigshospitalet, Copenhagen, Denmark. ⁹These authors contributed equally: Amalie D. Haue, Peter C. Holm ¹⁰These authors jointly supervised this work: Henning Bundgaard, Søren Brunak ✉e-mail: henning.bundgaard@regionh.dk; soren.brunak@cpr.ku.dk

