## [Transparent Peer Review file · Communications Medicine]

Subgrouping patients with ischemic heart disease by means of the Markov Cluster algorithm: A cohort study of 72,249 patients represented by 3,046 diagnoses

Corresponding Author: Professor Søren Brunak

Version 0:

Reviewer comments:

Reviewer #1

(Remarks to the Author)

I read with interest the manuscript titled: "Subgrouping patients with ischemic heart disease by means of the Markov Cluster algorithm: A cohort study of 72,249 patients represented by 3,046 diagnoses".

This article describes a study that used unsupervised clustering analysis to identify subgroups of patients with ischemic heart disease (IHD) who had different clinical outcomes. The study included 72,249 patients, and identified 31 distinct clusters. Some of these clusters had a higher risk of new ischemic events, while others had a lower risk. All of the clusters that were at increased risk of death from non-IHD causes were also associated with an increased risk of all-cause mortality. The authors also performed enrichment analysis to explore the phenotypic characteristics of the different clusters. They found that clusters enriched for diabetes and diabetes complications were associated with an increased risk of all three outcomes. In contrast, clusters that were not associated with an increased risk of secondary ischemic events were not characterized by polypharmacy. The authors conclude that unsupervised clustering analysis of patients with IHD based on multimorbidity can identify subgroups of patients with different clinical outcomes.

The article is well written in general and data are nicely presented, however, there are major issues to be addressed:

1. Ischemic heart disease (IHD) is known for its heterogeneous nature, with presentations ranging from symptomatic to asymptomatic.
 - a. This raises the possibility that many individuals with asymptomatic IHD may have been excluded from the cohort. The selection criteria, particularly in this subgroup, might have introduced a selection bias. Such selection bias may generate an important overvaluation of diseases typically associated with asymptomatic IHD such as CKD, T2DM and rheumatic diseases. To mitigate this bias, the authors could have employed stratified sampling or incorporated additional diagnostic criteria for asymptomatic IHD.
 - b. Asymptomatic individuals are often diagnosed after routine stress tests and subsequent coronary angiography (CAG) or coronary computed tomography angiography (CCTA), and typically show better prognosis compared to stable angina (SA) patients and those with prior acute coronary syndromes (ACS). In light of the available clinical treatment options for IHD such as lipid-lowering therapies and anti-inflammatory agents, the clinical course of individuals with asymptomatic IHD and SA are very distant from those with prior ACS. It is suggested that stable angina, asymptomatic individuals, and those with prior ACS be considered as distinct groups, and clustering should be performed only within the groups.
2. In the "Data sources and study population" section, the authors state that "To increase the positive predictive value of IHD diagnoses and temporally align included patients, we further required that patients had been subjected to CAG or CCTA." Can the authors provide more information on why these procedures were chosen as inclusion criteria? If a patient has performed CAG or CCTA and has no significant coronary artery lesion, was this patient included as a IHD case?
3. In the "Findings" section, the authors state that "18 clusters (35,982 patients) had higher or lower risk of death from non-IHD causes (12 and six clusters, respectively), and 23 clusters had a statistically significant higher or lower risk for all-cause mortality." It would be helpful if the authors could clarify whether there is any overlap between the clusters that had a higher risk of death from non-IHD causes and the clusters that had a higher risk of all-cause mortality.

4. In the "Enrichment analysis" section, the authors state that "We defined patients with polypharmacy as cases with more than five different recurrent prescriptions each redeemed two or more times within 1 year prior to the index date." Can the authors provide more information on why they chose this definition of polypharmacy?

5. In the "Discussion" section, the authors state that "The overall argument that motivated this study, is that graph theory and the application of unsupervised clustering to patient record data is a valuable approach to obtain a better understanding of IHD." Can the authors provide more information on how this study contributes to a better understanding of IHD? At this stage, the manuscript is clearly data-oriented, but it is unclear how it could be clinically helpful or how it could lead to a better understanding of IHD.

6. In the "Limitations" section, the authors state that "Three limitations of the study are ... (2) that the temporal order of the prior diseases was not considered in the clustering, ..." Can the authors provide more information on how this particular limitation might have affected the results of the study?

Reviewer #2

(Remarks to the Author)

The authors of this study have conducted a large retrospective analysis based on health record data (assuming electronic health records) accrued in health care of patients treated in Denmark. By using an unsupervised ML model, they have clustered the population identified to have ischemic heart disease by non-invasive or invasive imaging. They then have proceeded to provide association results for those clusters identified by the registry data. They provide association results for several phenotypic traits recorded around the date when IHD was identified, for polygenetic risk scores and for incident events after the "baseline".

The manuscript is well written, the graphics, tables and overall presentation is high quality work. The statistical analysis protocol seems rigorous. I commend the authors for their work.

There are several issues that reflect negatively on the publication value of this work that the authors could consider addressing.

1. There is no hypothesis that this work addresses. In the very end of the introduction section the authors list the things they aimed to do in this work, but the reader is left to guess, why or rather what for? In the abstract section the authors state: "We hypothesized that unsupervised clustering analysis could facilitate identification of distinct and clinically relevant multimorbidity clusters."

Please be more precise. What does this mean? What would one do with this cluster information? How are these classifications clinically relevant?

To be more precise, it is not impossible to assume that clustering using standardized ICD-10 format (or some other format) diagnostic data in a homogenous population such as that of predominantly Caucasian Danish population, could lead to fairly good results in population stratification in a manner that would yield statistically significant results in later association analyses. This fact does not require much proof anymore.

Markov modes or some other unsupervised learning method are a good way to regroup the population to different clusters. Proving that, does not seem to be the focus of this work either. Comparison of different clustering algorithms could have been performed even with simulated data but methodological comparison does not seem to be the point here either.

The authors do not provide metrics on how well their clustered data predict any of those outcomes they looked at. Finding associations in a dataset this large is not a problem one would assume, but showing that this overall process of using unfiltered standardized registry data could actually benefit clinical work flow in some way would be interesting indeed. However, it seems that that is not what the authors were after (?) Could there be some specific examples where this type of approach would be very effective.

2. Diagnosis data is recorded for many different purposes and not always to objectively reflect patients' prevalent conditions. Every physician knows that recording of this type of data is done for example for reimbursement purposes, to justify medical interventions, to justify other decisions and sometimes diagnostic data is not recorded at all because it simply leads to increased workload with no benefit to anybody. In other words, retrospectively collected data can include much more information than only objective data of prevalent conditions and given treatments that sure can improve clustering of patients. However, that information can be very context specific and does not translate to other systems/context. I.e. data recorded in Denmark might not be valuable for building models to be used in England or in China. Could the authors find some way to show that this method of clustering patients could be transferable to other countries / environments?

3. Using incident IHD event as an outcome is problematic. In supplementary material the authors have tried to define what counts as an incident event. Foremost problem is that there should be colors in that image but there are none in the submitted material. Second, 75% of all Mis as an incident event (in the supplementary material the authors have named this

as “Myocardial infarct as a secondary event”), have occurred within one month of the IHD index diagnosis. Almost similar graph is presented for revascularizations. While in cardiology it is true that often ad hoc PCI is done in the same session as the coronary angiography, it is unlikely that even a majority of these “secondary” events would be independent of the index diagnosis. Most likely they are scheduled elective revascularization procedures (complex PCI, CABG or just very simple PCI after discovery of CAD in coronary CT images). The same problem applies to the incident MI diagnoses. Either they are related to the index event or to revascularizations after discovery of CAD.

I would suggest that this endpoint be clarified in some way. Furthermore please provide information on the subdistributions of different components of this endpoint (what proportion died due to IHD, the proportion of “new” revascularizations and the proportion of new MIs”.

Other issues:

1. In section “Survival analysis” (Page 7 row 178) the authors state that there were three competing outcomes: “To investigate the association between cluster-membership and the competing risks of new ischemic events and death from non-IHD causes, and all-cause mortality, we used Cox proportional-hazards models (Cox models). For definitions of outcomes, see S0 Appendix.”

Why use Cox-regression analysis when competing events could be accounted for using Fine-Gray subdistribution analysis?

Supplementary Table 7 – Please explain what the effect size means and explain abbreviations in tables separately.

Version 1:

Reviewer comments:

Reviewer #1

(Remarks to the Author)

Thank you for your feedback on the revised manuscript. While I appreciate the substantial improvements, I still find the response to my second point inadequate. Specifically, I suppose the limitations section would be updated to incorporate the information discussed in the authors' response. Upon reviewing the revised manuscript, I cannot identify these changes. Could you please confirm whether the limitations section was updated and, if so, point me to the specific additions?

Reviewer #2

(Remarks to the Author)

The authors have addressed some of the issues raised. It is good to see that the clarification of what's a secondary event and what is classified as primary event in the survival analysis. Overall the methodology used in this work is good quality and the same can probably be said of the Danish registry data in general. The foremost limitation of this work is that it does not really show how this information is useful for the stakeholders in question. The efficacy of statistical modelling is unquestionable when using large datasets but what is the practical value of this information for the clinician or the patient.

Prediction approach was not equated with a diversity mapping effort. The authors have presented data of associations between identified clusters in a survival model and justify the importance of the results based on those results. Those results are presented right in the second sentence of the “Findings” paragraph of the abstract. How can one not, assume that the importance of clustering is NOT related to survival analysis (and prediction of new events) when this is one of the main themes of this work? I strongly recommend providing the reader some evidence of the actual value of this information rather than just show associations which are bound to exist in the data when nothing is truly recorded purely based on screening (agnostic recording of information). Practical examples would help.

Additionally, it would be very valuable to try to quantify in some way the proportionality of the inherent bias of recorded registry data. The comment was also meant to ask the question: How much of the ICD-10 data in Denmark comprises inherently biased information that reflect mostly data recording practices but does not provide much new information. A general practitioner most likely already knows their patient with multiple diagnoses is in elevated risk of IHD in the future and might also have higher genetic risk for CV conditions and worse metabolic profile. The relevant question is can the ICD-10 registry data improve above and beyond that knowledge?

First, using ICD-10 format data for clustering within a system and a population as homogeneous as the one in Denmark, does not mean that the results be repeatable anywhere else (perhaps besides some other Nordic country). The authors must have misunderstood the comment entirely. The replicability issues remain unanswered. One of the promise of “big data” is that models build on that data could be in some shape or form transferable.

Version 2:

Reviewer comments:

Reviewer #1

(Remarks to the Author)

I have no further comments

Reviewer #2

(Remarks to the Author)

Overall, this work represents good quality work in terms of its presentation, the analytical approach and the background material.

Despite now several rounds of peer review, many of the issues that were raised have not been addressed. Mainly the authors seem to have decided to not deal with them and keep the presentation as it is. Despite these fundamental differences in view points, publishing the observations of this manuscript may still provide valuable for the readership. However, I'd still advice the authors to reconsider their view that using ICD-10 format data even in a country like Denmark, would be free from bias. Therefore just sharing the code used in the analyses, does not guarantee reproducible results.

Reviewers' comments:

Reviewer #1 (Remarks to the Author):

I read with interest the manuscript titled: "Subgrouping patients with ischemic heart disease by means of the Markov Cluster algorithm: A cohort study of 72,249 patients represented by 3,046 diagnoses".

This article describes a study that used unsupervised clustering analysis to identify subgroups of patients with ischemic heart disease (IHD) who had different clinical outcomes. The study included 72,249 patients, and identified 31 distinct clusters. Some of these clusters had a higher risk of new ischemic events, while others had a lower risk. All of the clusters that were at increased risk of death from non-IHD causes were also associated with an increased risk of all-cause mortality. The authors also performed enrichment analysis to explore the phenotypic characteristics of the different clusters. They found that clusters enriched for diabetes and diabetes complications were associated with an increased risk of all three outcomes. In contrast, clusters that were not associated with an increased risk of secondary ischemic events were not characterized by polypharmacy. The authors conclude that unsupervised clustering analysis of patients with IHD based on multimorbidity can identify subgroups of patients with different clinical outcomes.

The article is well written in general and data are nicely presented, however, there are major issues to be addressed:

1. Ischemic heart disease (IHD) is known for its heterogeneous nature, with presentations ranging from symptomatic to asymptomatic.
 - a. This raises the possibility that many individuals with asymptomatic IHD may have been excluded from the cohort. The selection criteria, particularly in this subgroup, might have introduced a selection bias. Such selection bias may generate an important overvaluation of diseases typically associated with asymptomatic IHD such as CKD, T2DM and rheumatic diseases. To mitigate this bias, the authors could have employed stratified sampling or incorporated additional diagnostic criteria for asymptomatic IHD.
 - b. Asymptomatic individuals are often diagnosed after routine stress tests and subsequent coronary angiography (CAG) or coronary computed tomography angiography (CCTA), and typically show better prognosis compared to stable angina (SA) patients and those with prior acute coronary syndromes (ACS). In light of the available clinical treatment options for IHD such as lipid-lowering therapies and anti-inflammatory agents, the clinical course of individuals with asymptomatic IHD and SA are very distant from those with prior ACS. It is suggested that stable angina, asymptomatic individuals, and those with prior ACS be considered as distinct groups, and clustering should be performed only within the groups.

Response 1: We are glad that the reviewer appreciates this characteristic of IHD, as the wide range of symptoms was one of the things that prompted us to set out this study. However, we did not set out the study with a screening perspective and therefore asymptomatic IHD was beyond the scope, as explained in detail below.

Response 1a: We think the reviewer may have misunderstood the underlying data. These are routine care electronic patient records and not screening data. Therefore asymptomatic IHD is notoriously difficult to capture. Even if it was attempted, it would likely be a highly biased subgroup within asymptomatic IHD that could be added to the dataset. Further, we align the patients in time at either first CCTA or CAG fulfilling certain criteria. While we are aware that the onset of a chronic disease is non-trivial to define precisely, we explain in the Supplemental Material how this has been accounted for. Inclusion of asymptomatic IHD would not allow for a survival analysis either, as the time of diagnosis is unknown.

A stratified sampling approach would not align with the study design, as a stratified sampling approach requires certain information regarding the prespecified subgroups. As the reviewer states, IHD is a heterogeneous disease. Therefore we planned this study to assess if the MCL clustering could be applied to identify distinct patient subgroups at different risk of disease progression. In fact, our approach may provide information for future stratified sampling as e.g., some clusters are enriched for CKD (C30), T2DM (C5) and rheumatic diseases (C31).

Also, from the comments, it is not entirely clear if the reviewer by asymptomatic IHD means coronary atherosclerosis.

Changes made in the manuscript: In the Discussion, we have now included text on the potential importance of asymptomatic IHD, and also that it would be unfeasible to add this to the current study.

Response 1b: We have presented an unbiased, descriptive clustering of the cohort. We cannot completely follow the reviewer in combining the approach with a pre-determined subgrouping that also would reduce the power and the robustness of the clustering. While it was the goal of the present study to obtain a broad mapping of IHD, we agree that the approach could be applied to specific pre-determined subgroups that would allow for comparison of differences within these particular groups.

Changes made in the manuscript: In the Discussion, we have stressed the response to point 1b as to make it clearer that the clustering is data-driven and not formed by pre-defined subgroups.

2. In the "Data sources and study population" section, the authors state that "To increase the positive predictive value of IHD diagnoses and temporally align included patients, we further required that patients had been subjected to CAG or CCTA." Can the authors provide more information on why these procedures were chosen as inclusion criteria? If a patient has performed CAG or CCTA and has no significant coronary artery lesion, was this patient included as a IHD case?

Response 2: This is a crucial point regarding patient inclusion. We thank the Reviewer for asking us to clarify this aspect. The procedures were chosen as inclusion criteria to increase the likelihood that included patients were thoroughly examined for their IHD, i.e., an objective sign of IHD as demonstrated by imaging modalities. We did not include cardiac scintigraphy as this test is seldom used in the context of newly diagnosed IHD.

In Denmark, an ICD-code is no guarantee that the patient has been subjected to the proper diagnostics. Therefore, we required that patients had a relevant diagnosis code as well as a relevant diagnostic test and further assessed if the test was performed within reasonable distance in time to the diagnosis.

The assumption is that patients with no significant lesion on either CAG or CCTA, would not receive the IHD diagnosis. Therefore, a patient subjected to CAG or CCTA without a relevant IHD diagnosis would not be included. For example, if a patient was referred to CAG/CCTA with an IHD diagnosis, but the diagnosis was not re-assigned after the CAG/CCTA, the patient would not be included. The population-wide dataset used to form the cohort contains many patients with the IHD code and without the procedure, and vice versa.

Changes made in the manuscript:

3. In the "Findings" section, the authors state that "18 clusters (35,982 patients) had higher or lower risk of death from non-IHD causes (12 and six clusters, respectively), and 23 clusters had a statistically significant higher or lower risk for all-cause mortality." It would be helpful if the authors could clarify whether there is any overlap between the clusters that had a higher risk of death from non-IHD causes and the clusters that had a higher risk of all-cause mortality.

Response 3: We thank the Reviewer for bringing this point up. We find it highly relevant.

Changes made in the manuscript: We have calculated the overlap in terms of outcomes and added it to the "Findings" section. In replying to this criticism we further realized that we had introduced typos in Table 2 (one column), which have now been fixed.

4. In the "Enrichment analysis" section, the authors state that "We defined patients with polypharmacy as cases with more than five different recurrent prescriptions each redeemed two or more times within 1 year prior to the index date." Can the authors provide more information on why they chose this definition of polypharmacy?

Response 4: Polypharmacy is often defined as having five or more different drugs prescribed simultaneously, cf. Rochon et al., (2021) in *Lancet Healthy Longev*. One could have chosen other cutoffs and many patients receive indeed between 10 and 15 drugs. We aimed for making a broad mapping and would not focus on the most multimorbid patients only. Low risk patients which are currently overtreated are also a significant societal challenge. Therefore we chose a lower cutoff.

Changes made in the manuscript: We have included Rochon et al. as reference 49.

5. In the "Discussion" section, the authors state that "The overall argument that motivated this study, is that graph theory and the application of unsupervised clustering to patient record data is a valuable approach to obtain a better understanding of IHD." Can the authors provide more information on how this study

contributes to a better understanding of IHD? At this stage, the manuscript is clearly data-oriented, but it is unclear how it could be clinically helpful or how it could lead to a better understanding of IHD.

Response 5: We appreciate that the Reviewer acknowledges the data-oriented design, but also the relevant constructive critique. One example is that we clearly see that the mortality is not always driven by IHD, but rather by COPD, as the cluster enriched for COPD associates with an increased risk of death from non-IHD causes, but not new ischemic events. No matter how aggressively one would treat these patients for their IHD, it would likely be the comorbidity that would determine their survival.

Changes made in the manuscript: We have now elaborated on this aspect in the Discussion as requested.

6. In the "Limitations" section, the authors state that "Three limitations of the study are ... (2) that the temporal order of the prior diseases was not considered in the clustering, ..." Can the authors provide more information on how this particular limitation might have affected the results of the study?

Response 6: First of all, it is not easy to cluster disease trajectories as they have different lengths across different age groups. Many of the resulting clusters would be very small. Therefore it is not feasible in the current study setting. We already cite a paper that in detail analyses the order in which IHD patients develop diseases (ref. 14); both before and after the IHD diagnosis.

The clustering may in some cases be impacted by the order of the prior diseases and possibly the rate of disease progression, even if the diseases might be the same. However, we have now stressed that the fact that we align the patients at IHD diagnosis limits the impact of not incorporating the disease trajectories in full detail. We have taken great care in defining onset of IHD, but it would be close to impossible to do the same for all the other diseases given the data foundation.

Changes made in the manuscript: We have expanded the Discussion of this aspect of the analysis.

Reviewer #2 (Remarks to the Author):

The authors of this study have conducted a large retrospective analysis based on health record data (assuming electronic health records) accrued in health care of patients treated in Denmark. By using an unsupervised ML model, they have clustered the population identified to have ischemic heart disease by non-invasive or invasive imaging. They then have proceeded to provide association results for those clusters identified by the registry data. They provide association results for several phenotypic traits recorded around the date when IHD was identified, for polygenic risk scores and for incident events after the "baseline".

The manuscript is well written, the graphics, tables and overall presentation is high quality work. The statistical analysis protocol seems rigorous. I commend the authors

for their work.

There are several issues that reflect negatively on the publication value of this work that the authors could consider addressing.

1. There is no hypothesis that this work addresses. In the very end of the introduction section the authors list the things they aimed to do in this work, but the reader is left to guess, why or rather what for? In the abstract section the authors state: “We hypothesized that unsupervised clustering analysis could facilitate identification of distinct and clinically relevant multimorbidity clusters.”

Please be more precise. What does this mean? What would one do with this cluster information? How are these classifications clinically relevant?

Response 1a: We agree that the purpose of the study could be more precisely defined and thank the reviewer for pointing this out. Defining subgroups is an essential component in achieving precision medicine. A number of complex diseases like diabetes and heart diseases are very heterogenous, where a mapping of the diversity in a large, population-wide cohort is a useful basis for understanding the relationship between diversity and mortality.

If one wants to compare a new patient to the clusters, this can easily be done by including the patient in a new round of clustering or using a “patients-like-me-approach”. As an example in relation to clinical utility, we find that the mortality is not always driven by IHD, but rather by COPD, as the cluster enriched for COPD (C14) associates with an increased risk of death from non-IHD causes, but not new ischemic events (Table 2). No matter how aggressively one would treat these patients for IHD, it would likely be the comorbidity that would determine their survival.

Changes made in the manuscript: We have now clarified the purpose of the paper according to the remarks above in part by using COPD as an example.

To be more precise, it is not impossible to assume that clustering using standardized ICD-10 format (or some other format) diagnostic data in a homogenous population such as that of predominantly Caucasian Danish population, could lead to fairly good results in population stratification in a manner that would yield statistically significant results in later association analyses. This fact does not require much proof anymore.

Response 1b: We thank the reviewer for accepting that our population-wide approach is robust and warranted.

Changes made in the manuscript: None.

Markov modes or some other unsupervised learning method are a good way to regroup the population to different clusters. Proving that, does not seem to be the focus of this work either. Comparison of different clustering algorithms could have been performed even with simulated data but methodological comparison does not seem to be the point here either.

Response 1c: Our purpose is not to benchmark clustering algorithms, but to map the diversity in IHD comorbidity. All clustering algorithms have run-time dependencies and it is not feasible to get those run-time-parameters fully out of the equation. Minor differences in clustering results would likely not reflect IHD heterogeneity but rather algorithmic differences. By preparing the data taking IHD onset into account we, unlike others, have provided a much more robust starting point for the clustering.

Changes made in the manuscript: We have now stressed these aspects in the Discussion.

The authors do not provide metrics on how well their clustered data predict any of those outcomes they looked at. Finding associations in a dataset this large is not a problem one would assume, but showing that this overall process of using unfiltered standardized registry data could actually benefit clinical work flow in some way would be interesting indeed. However, it seems that that is not what the authors were after (?) Could there be some specific examples where this type of approach would be very effective.

Response 1d: The reviewer seems to equate a prediction approach with a diversity mapping effort. We are in other work studying the prognostic impact and time-to-event patterns, but in such designs the descriptive mapping remains a black box. The two approaches are complimentary and what we present here is a map and not a prognostication tool.

Changes made in the manuscript: None, as this was already mentioned in the Discussion.

2. Diagnosis data is recorded for many different purposes and not always to objectively reflect patients' prevalent conditions. Every physician knows that recording of this type of data is done for example for reimbursement purposes, to justify medical interventions, to justify other decisions and sometimes diagnostic data is not recorded at all because it simply leads to increased workload with no benefit to anybody. In other words, retrospectively collected data can include much more information than only objective data of prevalent conditions and given treatments that sure can improve clustering of patients. However, that information can be very context specific and does not translate to other systems/context. I.e. data recorded in Denmark might not be valuable for building models to be used in England or in China. Could the authors find some way to show that this method of clustering patients could be transferable to other countries / environments?

Response 2a: We acknowledge that generalisability is an important aspect of method development. However, the presented method is only based on ICD-10-codes and it is therefore easily transferable to almost all other settings. The quality of the Danish one-payer system based coding has been covered in many papers. As other papers cover the differences in coding practice across countries, we think this request is beyond the scope of the current study and that it would not add much beyond what is already known about the peculiarities of coding in different countries. While this is a relevant theme, it is essentially unrelated to the method we use.

Changes made in the manuscript: The code will be available upon publication for application of the method to data from other settings. This is already highlighted in the Code availability statement.

3. Using incident IHD event as an outcome is problematic. In supplementary material the authors have tried to define what counts as an incident event. Foremost problem is that there should be colors in that image but there are none in the submitted material.

Response 3a: Assuming we are here dealing with S1 Fig, we thank the Reviewer for paying careful attention to this and realize that the colors were somehow deleted in the process of uploading/submitting the material. In the uploaded doc-files (local copies) the colors are present. Possibly, the colors were lost during conversion to pdf.

Changes made in the manuscript: We have now uploaded a pdf-version of the supplemental material, where S1 Fig A and B are appropriately colored. Further we realized that there was an inconsistency in Figure 2, which have not been replaced by an updated version.

Second, 75% of all Mis as an incident event (in the supplementary material the authors have named this as “Myocardial infarct as a secondary event”), have occurred within one month of the IHD index diagnosis. Almost similar graph is presented for revascularizations. While in cardiology it is true that often ad hock PCI is done in the same session as the coronary angiography, it is unlikely that even a majority of these “secondary” events would be independent of the index diagnosis. Most likely they are scheduled elective revascularization procedures (complex PCI, CABG or just very simple PCI after discovery of CAD in coronary CT images). The same problem applies to the incident MI diagnoses. Either they are related to the index event or to revascularizations after discovery of CAD.

I would suggest that this endpoint be clarified in some way. Furthermore please provide information on the subdistributions of different components of this endpoint (what proportion died due to IHD, the proportion of “new” revascularizations and the proportion of new Mis”.

Response 3b: It seems that our presentation has led the Reviewer to misunderstand S1 Fig and agree with the Reviewer that this is an important aspect of the end-point definition. Therefore, we had made and presented S1 Fig. In a data-driven manner, we argued that events considered secondary ischemic events most likely are independent of the index diagnosis. We have applied the “elbow-method” to show that 75% of revascularizations have occurred within 90 days of the index event, and therefore, these procedures are not considered secondary ischemic events. Similarly, 75% of subsequent hospitalizations for MI have occurred within the first 30 days of the index event and are therefore not included in the end-point. The unifying argument is that planned revascularization after the discovery of CAD follow similar temporal patterns. We think our handling of this difficult aspect is quite unique in the literature and hope that the Reviewer will acknowledge that.

The information regarding subdistributions of the endpoint were already included in Table 1.

Changes made in the manuscript: We have now relocated the definitions of the outcomes in the survival analysis from the supplemental material to the main manuscript (Methods) and mentioned the potential impact of the 25% of cases that are included as an outcome, although some of them might be related to the index event.

Other issues:

1. In section “Survival analysis” (Page 7 row 178) the authors state that there were three competing outcomes: “To investigate the association between cluster-membership and the competing risks of new ischemic events and death from non-IHD causes, and all-cause mortality, we used Cox proportional-hazards models (Cox models). For definitions of outcomes, see S0 Appendix.”

Why use Cox-regression analysis when competing events could be accounted for using Fine-Gray subdistribution analysis?

Response 1: We decided to use Cox-regression analysis since this is the most widely applied survival method. Moreover, we would like to refer to Andersen et al (2012) (cf. <https://doi.org/10.1093/ije/dyr213>) where it is stated that “Cause-specific hazards may be more relevant when the disease aetiology is of interest, since it quantifies the event rate among the ones at risk of developing the event of interest. Cumulative incidences are easier to interpret and are more relevant for the purpose of prediction.” As our paper deals with the aetiology of IHD in the context of multimorbidity, we argue that the ideal application of the respective methods is not clear-cut.

Changes made in the manuscript: We have added the suggestion from the Reviewer to the Discussion, but also mentioned that it would have higher relevance in the context of prediction.

Supplementary Table 7 – Please explain what the effect size means and explain abbreviations in tables separately.

Response: The effect size reflects the result of the Wilcoxon rank-sum test described on page 8.

Changes made in the manuscript: We have added an extensive legend to the table to make the explanation above and any abbreviations clearer to the reader.

Reviewers' comments:

Reviewer #1 (Remarks to the Author):

Thank you for your feedback on the revised manuscript. While I appreciate the substantial improvements, I still find the response to my second point inadequate.

Response: We appreciate the Reviewer acknowledged the improvements and address the second point below.

Changes in the manuscript: See below.

Specifically, I suppose the limitations section would be updated to incorporate the information discussed in the authors' response. Upon reviewing the revised manuscript, I cannot identify these changes. Could you please confirm whether the limitations section was updated and, if so, point me to the specific additions?

Response: The first submitted manuscript included three limitations. To address the important limitation raised by the Reviewer we expanded this section to include a fourth limitation, cf.:

“ (...), and (4) the design does not allow for inclusion of asymptomatic IHD patients.” (page 17, bottom of top paragraph)

In the previous revision we also included the Fine-Gray subdistribution analysis as an alternative and argued that Cox modelling is a superior choice of method in the current design (cf. p. 17; bottom of top section).

With the present revision we have further expanded on the limitation section to address the points raised by Reviewer #2. I.e., we have highlighted that diseases are not always diagnosed in the order in which they evolve which is a source of noise in the data.

Changes in the manuscript: To the limitations section we have now added

“ (...), and (5) quality of the coding.” (page 17, bottom of top paragraph)

“The fifth limitation concerns the quality of the coding that often is debated across settings. It is crucial to be aware of the fact, that the clustering was performed on secondary care data. All patients had their diagnosis of IHD confirmed either by CAG or CCTA. The ICD-10 codes from Denmark used as input for the MCL algorithm have been extensively validated in the literature^{26,30,31}.” (bottom of second paragraph pp. 17 and 18).

Reviewer #2 (Remarks to the Author):

The authors have addressed some of the issues raised. It is good to see that the

clarification of what's a secondary event and what is classified as primary event in the survival analysis. Overall the methodology used in this work is good quality and the same can probably be said of the Danish registry data in general.

The foremost limitation of this work is that it does not really show how this information is useful for the stakeholders in question. The efficacy of statistical modelling is unquestionable when using large datasets but what is the practical value of this information for the clinician or the patient.

Response: Here, we would like to stress that the goal of the study was not to create guideline-like information relevant for the individual patient, per se. Rather the overall goal is etiological and was to characterize specific patient subgroups that make up this heterogeneous population. As the study clearly demonstrates, there are differences with respect to risk of secondary ischemic events as well as other clinical outcomes among these subgroups. We argue that this information is indeed of relevance for clinical workup and precision medicine. Moreover, it may help inform future RCTs that could involve designs that take the subgroups into account to test whether differential treatment strategies in these subgroups can be shown to improve clinical outcomes. We would also like to draw the Reviewers attention to the recent "Commission on rethinking coronary artery disease: moving from ischaemia to atheroma" (Zaman *et al.* Lancet, 2025) that aligns very well with the overall argument of our study.

Changes in the mansscript: We have now further added these translational perspectives to the Discussion (cf. bottom of the second paragraph pp. 15 and 16). Also, we have added a citation to The Lancet review mentioned above.

51. Zaman, S. *et al.* The Lancet Commission on rethinking coronary artery disease: moving from ischaemia to atheroma. *The Lancet* 405, 1264–1312 (2025).

Prediction approach was not equated with a diversity mapping effort. The authors have presented data of associations between identified clusters in a survival model and justify the importance of the results based on those results. Those results are presented right in the second sentence of the "Findings" paragraph of the abstract. How can one not, assume that the importance of clustering is NOT related to survival analysis (and prediction of new events) when this is one of the main themes of this work? I strongly recommend providing the reader some evidence of the actual value of this information rather that just show associations which are bound to exist in the data when nothing is truly recorded purely based on screening (agnostic recording of information). Practical examples would help.

Response: We are glad that the reviewer raises this point and have expanded on the "Findings" section based on this input. This argumentation is again based on the assumption of the reviewer that we aimed for prediction of future events, which was not the case (as already mentioned above).

The relationship between clustering and survival is a main result as the input data in an unsupervised approach end up leading to a survival stratification that is illustrative of the multimorbidity of IHD patients. Again, we are not predicting the next event; we are grouping IHD patients based on their past and demonstrate that this principle of grouping nicely correlates with progression patterns as modeled by the Cox proportional hazards method. We therefore find that adding examples of how the work leads to predictions, which are relevant clinically, is out of scope. We already in the previous rebuttal mentioned that a machine learning algorithm would be much better suited for predicting the occurrence of new events.

Changes in the manuscript: We have now added a reiteration to the Discussion to emphasize the etiological purpose over predicting new events (cf. bottom of second paragraph p. 16).

Additionally, it would be very valuable to try to quantify in some way the proportionality of the inherent bias of recorded registry data. The comment was also meant to ask the question: How much of the ICD-10 data in Denmark comprises inherently biased information that reflect mostly data recording practices but does not provide much new information. A general practitioner most likely already knows their patient with multiple diagnoses is in elevated risk of IHD in the future and might also have higher genetical risk for CV conditions and worse metabolic profile. The relevant question is can the ICD-10 registry data improve above and beyond that knowledge?

Response: The reviewer is likely unaware of the underlying data structure. Owing to the nationwide structure of The Danish National Patient Registry the inherent bias that the Reviewer alludes to is not present. The cohort is selected from the entire country and in that sense it is not biased towards any demographic subgroups, age groups or similar relative to the underlying population. The study was in fact set out to answer the question asked by the Reviewer in the last sentence.

While the question regarding a quantification of how much the ICD-10 codes reflect recording practices rather than natural courses of diseases is a key question and we are glad that the Reviewer highlights this feature of the study. Explicitly, it is a major strength that only limited filters were applied prior to clustering. It is crucial to be aware of the fact that the clustering was performed on secondary care data. I.e., the importance of the memory of a general practitioner does not make much sense in this context (all patients had their diagnosis of IHD confirmed by either coronary arteriography (CAG) or chest coronary computed tomography angiography (CCTA), cf. Methods first paragraph on p. 4).

The ICD-10 codes used as input for the MCL algorithm have been extensively validated in the literature, cf. refs. #26.: Schmidt et al. (2019), #30.: Schmidt et al. (2014), and ref. #31.: Sundbøll et al. (2016) (inserted below).

26. Schmidt, M. et al. The Danish health care system and epidemiological research: from health care contacts to database records. *Clin. Epidemiol.* 11, 563–591 (2019).

30. Schmidt, M., Pedersen, L. & Sørensen, H. T. The Danish Civil Registration System as a tool in epidemiology. *Eur. J. Epidemiol.* 29, 541–549 (2014).

31. Sundbøll, J. et al. Positive predictive value of cardiovascular diagnoses in the Danish National Patient Registry: a validation study. *BMJ Open* 6, (2016).

Changes in the mansscript: We have now stressed these aspects further in the Discussion, cf. our second response to Reviewer #2 and in manuscript added text at the bottom of second paragraph pp. 17 and 18).

First, using ICD-10 format data for clustering within a system and a population as homogeneous as the one in Denmark, does not mean that the results be repeatable anywhere else (perhaps besides some other Nordic country). The authors must have misunderstood the comment entirely. The replicability issues remain unanswered. One of the promise of "big data" is that models build on that data could be in some shape or form transferable.

Response: Here, we will again mention that the code is ready and will be made publically available upon publication, which means that it can be used in the study of other data in different countries and settings. Please also note that trajectories of ICD codes (the input data) are available electronically in a very large number of countries. Thus, the generalisability of the study lies within the option of running the software locally. To this end, we have now with the revision already prepared the code for general access.

The repository contained the code that was developed for this study is available to download and install via <http://www.zenodo.org/> (URL) with DOI: 10.5281/zenodo.15211110.

Changes in the mansscript: We have now provided detailed instructions for how to obtain the software in the Code availability statement section.

Dear Dr. Lauren Malave,

Thank you very much for the e-mail and for the request to make a response to the remaining issue. We have updated the manuscript accordingly. Please find our responses to the Reviewers in blue below. We have also gone through the editorial checklist and made a large number of changes to adhere to the requirements of Communications Medicine. We have not listed those below, but have commented on each of them in the checklist.

Looking forward to hear from you.

Søren Brunak

Response to the reviews

Reviewer #1 (Remarks to the Author):

I have no further comments

Response: We thank the Reviewer for finding the revised paper satisfactory.

Changes made in the manuscript: None.

Reviewer #2 (Remarks to the Author):

Overall, this work represents good quality work in terms of its presentation, the analytical approach and the background material.

Despite now several rounds of peer review, many of the issues that were raised have not been addressed. Mainly the authors seem to have decided to not deal with them and keep the presentation as it is. Despite these fundamental differences in view points, publishing the observations of this manuscript may still provide valuable for the readership. However, I'd still advice the authors to reconsider their view that using ICD-10 format data even in a country like Denmark, would be free from bias. Therefore just sharing the code used in the analyses, does not guarantee reproducible results.

Response: We think that we in each case have argued why certain points do not need to be revised. We agree with the reviewer that the limitations around

ICD-10 format data can be be stressed additionally and have therefore followed the suggestion completely.

Changes made in the manuscript: We have revised the Discussion accordingly (cf. pp. 22-23 in the clean version of the revised manuscript).